# High entropy engineered polymer blends with enhanced dielectric properties and high temperature stability

Xin Qi[1], Xuankai Huang[2], Nasima Kanwal[2], Bijoy Das[1], Anthony E. Phillips[2], Dimitrios G. Papageorgiou[1], Haixue Yan[1], Emiliano Bilotti[3] & Michael J. Reece[1] ✉

There is increasing need for higher performance dielectric polymers for devices in power conversion systems for renewable energy generation and electric vehicles. In particular, materials with higher dielectric permittivity, lower loss and the ability to operate at higher temperatures. We have developed a counter intuitive method to achieve this, the melt blending of multiple immiscible polymers, in an approach that mimics high entropy materials design. We demonstrate that using this approach we can significantly exceed the rule-of-mixtures for the dielectric constant (>250%), whilst surprisingly retaining a low loss tangent. The materials show increased thermal stability up to 150 °C, which opens up the possibility of the wider application of dielectric polymers. We provide a consistent model to describe the behaviour based on the use of polymers with different glass transition temperatures to frustrate the de-blending of the immiscible polymers during melt processing. This produces highly amorphous and disordered polymer blends with increased inter-chain spacing (free volume) and increased rotational freedom of the polar groups in polar nano regions. This approach has wide applicability to other polar polymer blends and is scalable.

Dielectric devices are used in power conversion systems for renewable energy generation and electric vehicles, and they are currently predominantly based on ceramic materials[1]. Polymer and polymer composite dielectrics are of increasing interest because of their ease of processibility into films, low density, flexibility, low cost and high breakdown strength[2–4]. The origin of the high permittivity of current polymer based dielectrics is associated with the rotation of the dipole moment of polar side groups in the amorphous phase[5,6], and their switching in the polar crystalline phase in the case of ferroelectric polymers[3]. However, the relatively low melting temperature of polymers gives rise to relatively low thermal stability (<100 °C)[7,8]. A number of different approaches have been explored to produce thermally stable, high permittivity and low loss materials, and are well reviewed by Zhu[3]. A common approach to achieve higher performance is the engineering of hybrid composites and their interfaces. Significant improvements can come through optimising the contribution from the orientation polarisation mechanism by engineering dipole rotation and interactions, including through manipulating the interchain distance[4].

The highest dielectric permittivity and best ferroelectric polymer materials are fluoropolymers[9–11] with highly electronegative fluorine atoms[2,3,12,13]. However, these Per-/Poly-fluoroalkyl Substances (PFAS)[14–16] have a long term detrimental effect on the environment, and the compounds they produce when they breakdown are highly toxic and persistent[17,18]. In this work we present a counter intuitive approach to achieve high and stable dielectric permittivity materials, the non-equilibrium blending of immiscible polymers by easily scalable melt processing. While we demonstrate this approach using a model system

[1]School of Engineering and Materials Science, Queen Mary University of London, London, UK. [2]School of Physical and Chemical Sciences, Queen Mary University of London, London, UK. [3]Department of Aeronautics, Imperial College London, London, UK. ✉e-mail: m.j.reece@qmul.ac.uk

**Table 1 | Physical characteristics of the blended and individual polymers**

| Name | Molecular composition(mol%) | Molecular weights (kg mol⁻¹) | Fusion enthalpy for 100% crystalline (J g⁻¹) | Crystallinity (%) [Fig. 2] | Melting temperature (°C) [Fig. 2] | Glass transition temperature (°C) at 1 kHz [Figs. 2 and 5] |
|---|---|---|---|---|---|---|
| HEP-B1 | N/A | N/A | N/A | N/A | N/A | −9 |
| HEP-B2 | N/A | N/A | N/A | N/A | N/A | −38 |
| PVDF_6020 | 100 | 670–700 ($M_w$) (Solef®6020) | 104.7[30,31] | 38 | 171.3 | −13 |
| P(VDF-TrFE)_25 | 75/25 | N/A | 38[32] | 76 | 150.6 | −2 |
| P(VDF-HFP) | 96/4 | 400 ($M_w$) and 130 ($M_n$) | 104.6[33] | 27 | 147.0 | −18 |
| PP | 100 | 340 | 207.9[34] | 40 | 165 | −20 to −10[35] |
| PS | 100 | 200 | N/A | N/A | N/A | 100 |

containing fluorinated and non-fluorinated polymers, the same approach can be applied to all non-fluorinated polar polymer blends. This work was stimulated by the results of research previously reported on the blending of polyvinylidene fluoride (PVDF) and poly(-vinylidene fluoride-co-trifluoroethylene) (P(VDF-TrFE)) films prepared by melt extrusion[19–21]. The dielectric constant of the two blended films in similar proportions was 22.5 at 1 kHz, which exceeded the rule-of-mixtures (RoM) of the components by more than 50%[19,20]. This observation could not simply be explained by differences in crystallinity or preferred orientation of the films.

The aim for the current research was to investigate the possibility to further enhance the dielectric properties of polymers by the blending of multiple immiscible but co-processable polymers. The use of multiple polymer components mimics entropy engineering in metal alloys[22] and more recently in ceramics[23], increasing their mixing and disorder. The five polymers, namely PVDF, P(VDF-TrFE), poly(vinylidene fluoride - co-hexafluoropropylene) (P(VDF-HFP)), polypropylene (PP), polystyrene (PS), were chosen because they have similar melting temperatures, in the range of -150–170 °C (see Table 1), and a melt processing temperature of 200 °C was chosen. The fluorinated polymers are commonly used functional polymers that are highly polar, while PP and PS are inexpensive, commonly used and non-fluorinated polymers. In the current work, we have chosen a polymer blend of five immiscible polymers, four of which are semi-crystalline and have similar $T_g$ (−0 to −20 °C at 1 kHz) and the fifth component PS, is amorphous and has a much higher $T_g$ (-100 °C), referred to as Blend-1 (B1). The rationale for this approach is that the melt blending of the co-processable high entropy polymer blend in B1 would produce intimate blending of the individual polymers in the melt, which would frustrate the de-blending and crystallisation of the individual polymers during cooling (illustrated in Fig. 1a). The rationale for adding the PS to B1 was to increase the viscosity of the melt during cooling, further frustrating the de-blending of the polymers on cooling. To test our rationale, a second polymer blend was prepared with four of the same polymers, but without the addition of PS, Blend-2 (B2).

## Results

### Room temperature dielectric properties

High-quality films were produced by melt-extrusion, as evidenced by the uniform and translucent appearance of the individual polymers (Supplementary Fig. S1). The blended polymers, B1 and B2, are less transparent. The samples had a thickness of -150 μm. Figure 1b shows the room temperature dielectric properties of films of the two different polymer blends, B1 (PVDF, P(VDF-TrFE), P(VDF-HFP), PP, PS) and B2 (PVDF, P(VDF-TrFE), P(VDF-HFP), PP), and the individual polymer components processed under the same conditions. For the B1 polymer blend, the dielectric constant is 22.4 at 1 kHz, which is >250% higher than that estimated by a simple RoM of the weighted average of the

individual polymers (8.3 at 1 kHz). While for the B2 polymer blend, without the addition of PS, the dielectric constant of the multicomponent polymer is 8.7 at 1 kHz, which is close to that estimated by the RoM (9.6 at 1 kHz). The dielectric loss tangent for both of the polymer blends is relatively low (<0.05 at 1 kHz). In the case of the B2 polymer blend, its dielectric constant and loss tangent show similar frequency dependence to that of the individual fluorinated polymers, with a peak in the loss in the frequency range of 1–10 MHz. However, the B1 polymer blend shows a completely different behaviour, with an incipient loss peak at a frequency >100 MHz, beyond the frequency range of our equipment. These results clearly demonstrate that the melt blending of immiscible equi-proportion polymers leads to some degree of interaction between the polymers, that leads to structures with enhanced dielectric response, and not simply a de-blending of the individual polymers. Figure 1c shows the effect of reducing PS mass% content on the B1 polymer blend, with a consistent trend in the dielectric constant and loss tangent behaviour between those of the B1 and B2 blends. This behaviour can be explained by an increasing confinement of the active orientation polar regions with increasing PS content into smaller polar nano regions. It can be seen in Fig. 1c that at about 15 mass% PS content, the enhancement of dielectric constant saturates, and the effect of PS on the dielectric properties begins to decrease. Once the beneficial effects of PS addition are optimised, further incorporation results in a dilution of the dielectric constant of the B1 blend. It should also be noted that in previously reported work on a two-polymer blend of PVDF and P(VDF-TrFE)[19], a modest enhancement (-50%) of the dielectric constant was observed compared the expected value from the RoM. However, there was no noticeable change in the relaxation frequency. This relaxation behaviour is also consistent with the results for the B2 blend, which includes the three fluorinated polymers, along with PP, but excludes PS. The relaxation frequency of the B2 blend, is similar to that of the individual polymers. These observations highlight the critical role of PS in enhancing the dielectric properties of the B1 blend, and in reducing its relaxation time.

There are different polarization mechanisms in polymers, including interfacial, orientation (dipolar), ionic and electronic, from low to high frequency, depending on their relaxation frequency[3,4,24]. According to its relaxation frequency range (-100 MHz), the enhanced dielectric constant of the B1 blend can be attributed to the contribution of dipolar polarization, which is mostly produced by the PVDF based non-linear polar dielectric polymers[3,24]. Both B1 and B2 polymer blends include PVDF based polymers, however, the dielectric constant of B1 is much higher than that of B2 (note that B2 actually contains a higher proportion of the highly polar polymers than B1). As we will show, the high permittivity of B1 can be attributed to the increased rotation space (free volume) for dipolar polarization induced by the addition of PS, which is confirmed in the characterisation section by the XRD data in Fig. 2, with details in the next section[4,25]. Therefore, the

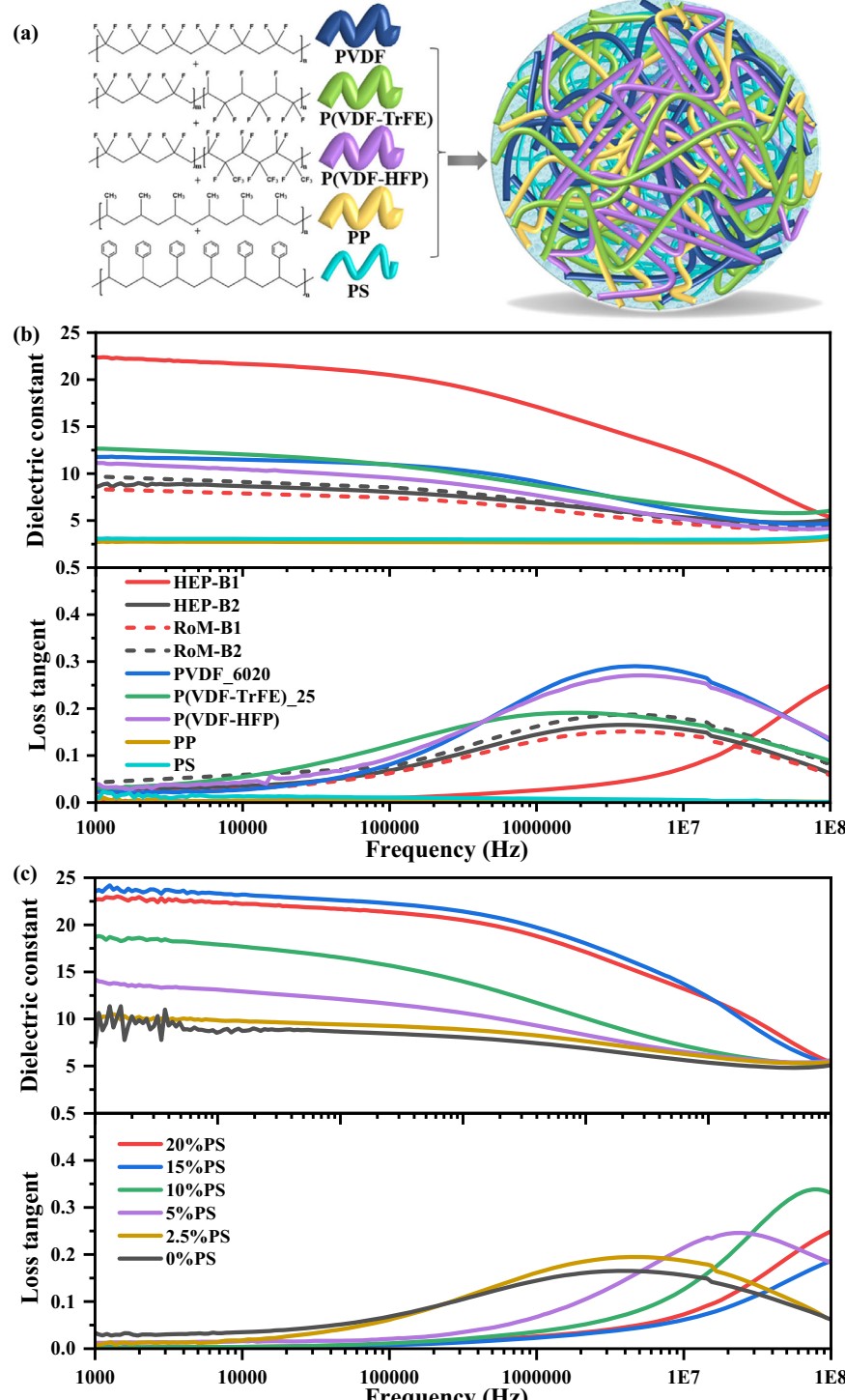

**Fig. 1 | Dielectric properties of HEP-B1 and HEP-B2 polymer blends and individual polymers.** a Molecular structures of the individual components used in HEP-B1 and HEP-B2, including PVDF, P(VDF-TrFE), P(VDF-HFP), PP, and PS, are depicted alongside a conceptual illustration of the blended polymer matrix. **b** Frequency-dependent dielectric constant and loss tangent of HEP-B1, HEP-B2, and their components, compared with the rule-of-mixture (RoM) predictions (RoM-B1 and RoM-B2). **c** Influence of varying PS content on dielectric properties in polymer blends.

dipolar polarization is due to polar nano regions (PNRs) associated with polar PVDF based polymers[3,4]. The dielectric relaxation frequency of the B1 blend is significantly higher than that of B2 blend, which can be attributed to reduced size of PNRs in the B1 blend. In the next sections the molecular structure of the polymers is investigated to elucidate the mechanisms responsible for the enhanced dielectric properties.

## Microstructure, phases and crystallinity

The DSC heating and cooling data for the individual polymers and the polymer blends is presented in Fig. 2a–d. The melt, crystallisation and glass transition ($T_g$) temperatures of the individual polymers, and the Curie Transition of P(VDF-TrFE) are consistent with the literature (Table 1). The polymer blends display the overlapping melting peaks of the PP, PVDF and P(VDF-TrFE). On the first heating (Fig. 2a) the Curie

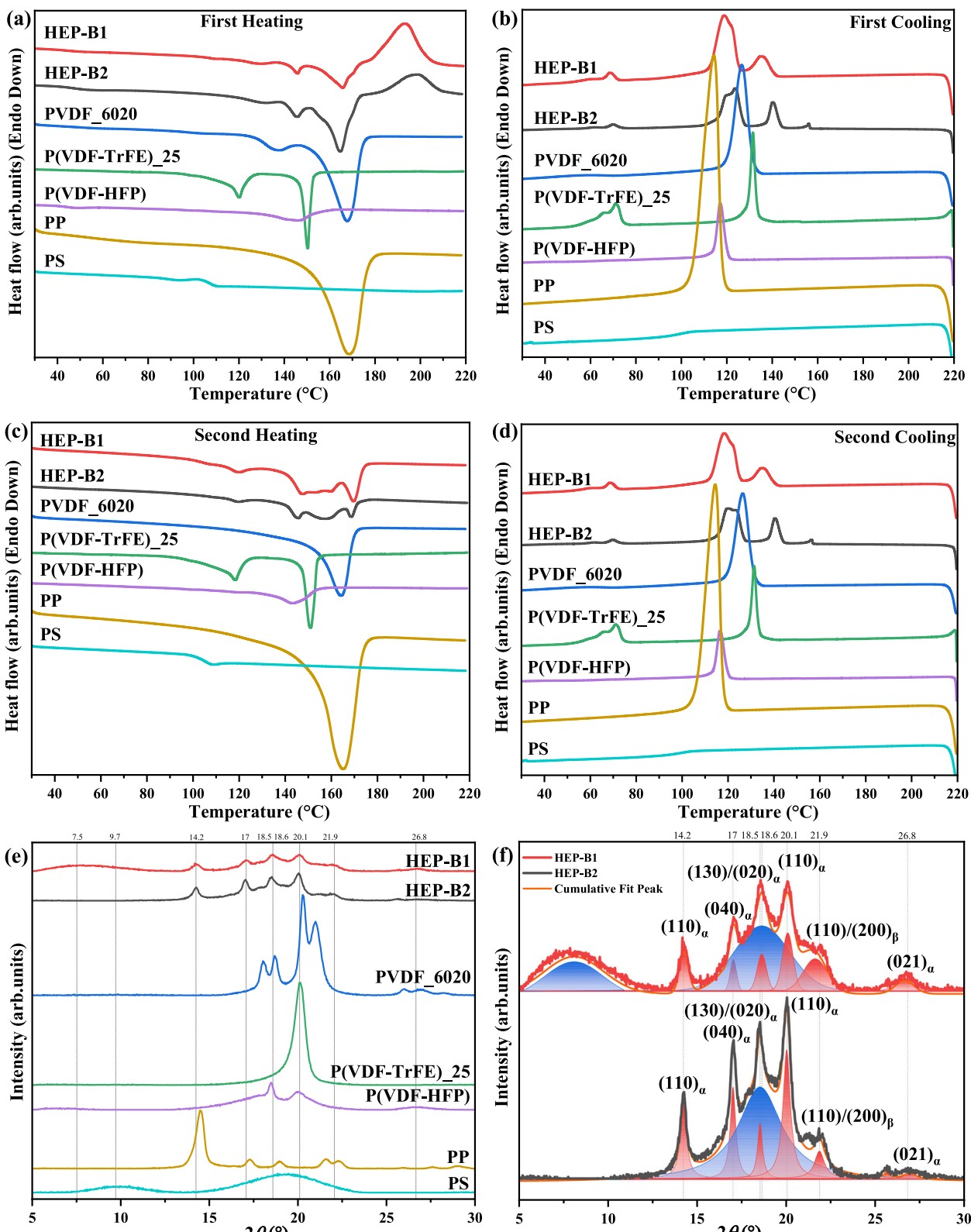

**Fig. 2 | Differential Scanning Calorimetry (DSC) and X-Ray Diffraction (XRD) analyses of the HEP-B1 and HEP-B2 polymer blends and individual polymers. a** DSC first heating curves. **b** DSC first cooling curves. **c** Second heating. **d** Second cooling. **e** XRD diffraction patterns of HEP-B1, HEP-B2, and individual polymers. **f** Enlarged XRD spectra, comparison between HEP-B1 and HEP-B2, showing peak fitting of amorphous and crystalline phases.

transition peak of P(VDF-TrFE) at ~120 °C is not apparent in either of the blends, because of a low content (<5%) of the crystalline ferro-electric β- phase. An additional significant and broad exothermic peak is present at about ~195 °C. This latter peak is likely to be related to a non-equilibrium structure, probably involving the partial blending of immiscible polymer components, which were kinetically trapped during cooling during melt extrusion. On first cooling (Fig. 2b), the DSC data of the blended polymers now looks more like a convolution

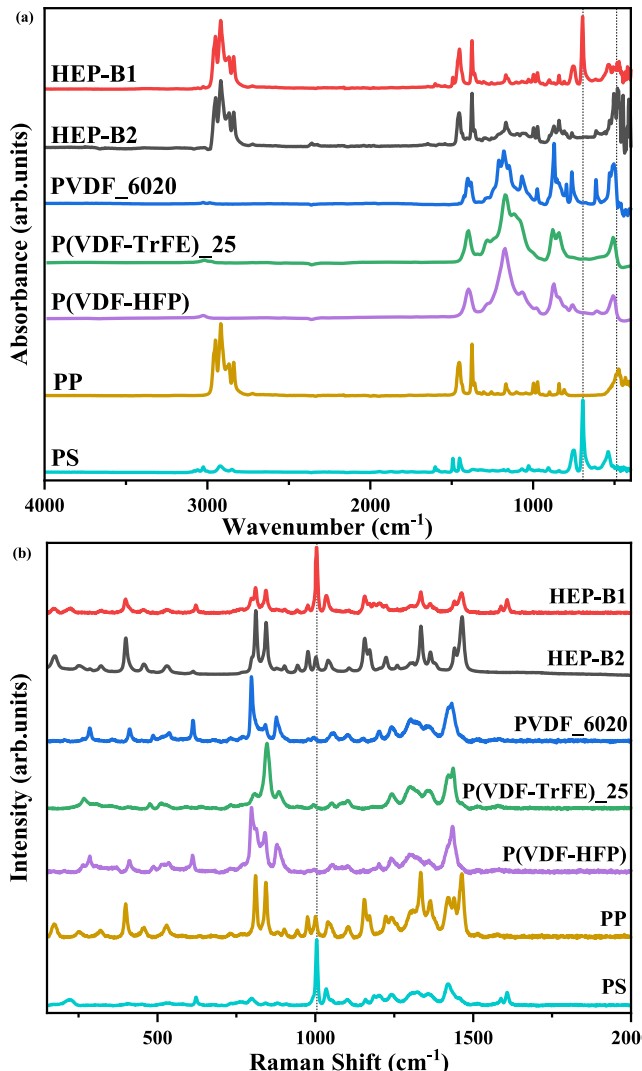

**Fig. 3 | Spectroscopic analysis of HEP-B1 and HEP-B2 polymer blends and individual polymers. a** FTIR spectra. **b** Raman spectra.

(110)$_\alpha$ peak of PP at ~14.5° has moved to a slightly smaller angle of 14.2° in the blended polymers.

The above observations can be explained as follows, on cooling, PS has a glass transition temperature at ~100 °C, so effectively loses macromolecular segmental mobility (i.e. becomes glassy) well before the amorphous components of the other polymers in the B1 blend, significantly increasing the viscosity of the cooling blended melt. The amorphous components of the other polymers would then be expected to solidify as if in the B2 blend, but within the skeletal matrix of the already frozen-in PS. There is also the possibility of some degree of miscibility or blending between the polymers in these two regions due to kinetic entrapment. This could play a significant role in determining the resulting structures, leading to more interfaces and the formation of polar nano regions with reduced size. Consequently, this structural evolution strongly influences the dielectric properties[3]. All of this is supported by the high-temperature exothermic transition (~195 °C) observed in the first heating DSC data, the increased amorphous phase content, the significantly enhanced dielectric constant and increased relaxation frequency of the B1 blend compared to the B2 blend. The higher dielectric permittivity of B1 can be attributed to the increased rotational freedom (free volume) of the polar nano regions, facilitating greater dipole mobility and polarization.

### Vibrational spectroscopy

In order to provide further evidence for the phases present and insight into the local environments of the polymer units, Fourier transform infrared (FTIR) spectra were collected. Figure 3a shows representative sweep data for all of the polymers and blends, with additional data presented in more detail in Supplementary Fig. S2 in the Supplementary Information. Figure 3b shows the corresponding Raman data (see Supplementary Fig. S3 in Supplementary Information for more detail). Analysis from different regions of the blended films showed some variation, reflecting local inhomogeneity at the scale of the probe sizes, ~5 mm for FTIR and 10–100 μm for Raman. However, no distinct difference between the blends and the individual polymers was found; the spectra of the blends were simply a convolution of the signal from the individual polymers. Considering that the enhanced dielectric response is observed at frequencies below $10^8$ Hz (Fig. 1b), vibrational spectroscopy, corresponding to much higher frequencies, is unable to provide direct evidence for the mechanism responsible for the enhanced dielectric response. This could explain why the blending behaviour reported here has not been reported previously using conventional vibrational spectroscopic techniques[19,20].

### Solid-NMR

To gain some insight into the environment of the polymers in the blends, solid-NMR analysis was performed. Figure 4a–c shows the NMR $^1$H, $^1$H-$^{13}$C, and $^{19}$F resonance data for the individual polymers and the B1 and B2 blends. The $^1$H and $^1$H-$^{13}$C spectra of the B1 and B2 blends simply look like a convolution of the peaks of the individual polymers. The additional peaks marked with an * are artifacts from the sample spinning. The $^{19}$F data is particularly relevant to study because the fluorinated polymers provide the largest contribution to the orientation polarisation of the blended polymers. The B1 and B2 spectra have all of the expected peaks associated with the different local environments in the individual PVDF-based polymers, corresponding to the α- and β-crystalline phases, and the amorphous phases, which suggests that the individual polymers have to some extent separated into their usual local environments[26]. All of this suggests that the higher dielectric response of the B1 blended polymers is not produced by a significant change in the local environments of its individual polymers, but is a consequence of the higher mobility and free volume of the fluorinated polymers in smaller polar nano regions.

of the data of the individual polymers. Additionally, no endothermic peak appears corresponding to the exothermic peaks seen at ~195 °C on first heating, and the Curie transition peak of P(VDF-TrFE) is now apparent at ~70 °C, indicative of the crystallisation of the polar β-phase. The second heating and cooling DSC data for the blended films simply looks like a convolution of the individual polymers, indicating that the blends have equilibrated into the individual polymer components.

The expected crystalline phases of the individual polymers were identified in the XRD patterns (Fig. 2e, f). The blended films are highly amorphous, as evidenced by the broad amorphous halo. It should also be noted that all of the XRD patterns were collected under the same conditions, and the intensity of the patterns of the blended films is relatively low because of their higher amorphous content. The degree of crystallinity of the blended films, determined from the fitting of the XRD data in Fig. 2f, is estimated to be 18% and 38% for the B1 and B2 films, respectively. As a crude comparison, the rule-of-mixtures average for the individual polymers is 36.2% and 45.3%, based on the data in Table 1. Additionally, note that the amorphous peak of PS at 9.7° has shifted to a lower angle of 7.5°, corresponding to an increased average intermolecular spacing in PS from 0.9 nm to 1.2 nm. These observations suggest that the PS in B1 is structurally very different to that of the pure polymer. Additionally, note that the

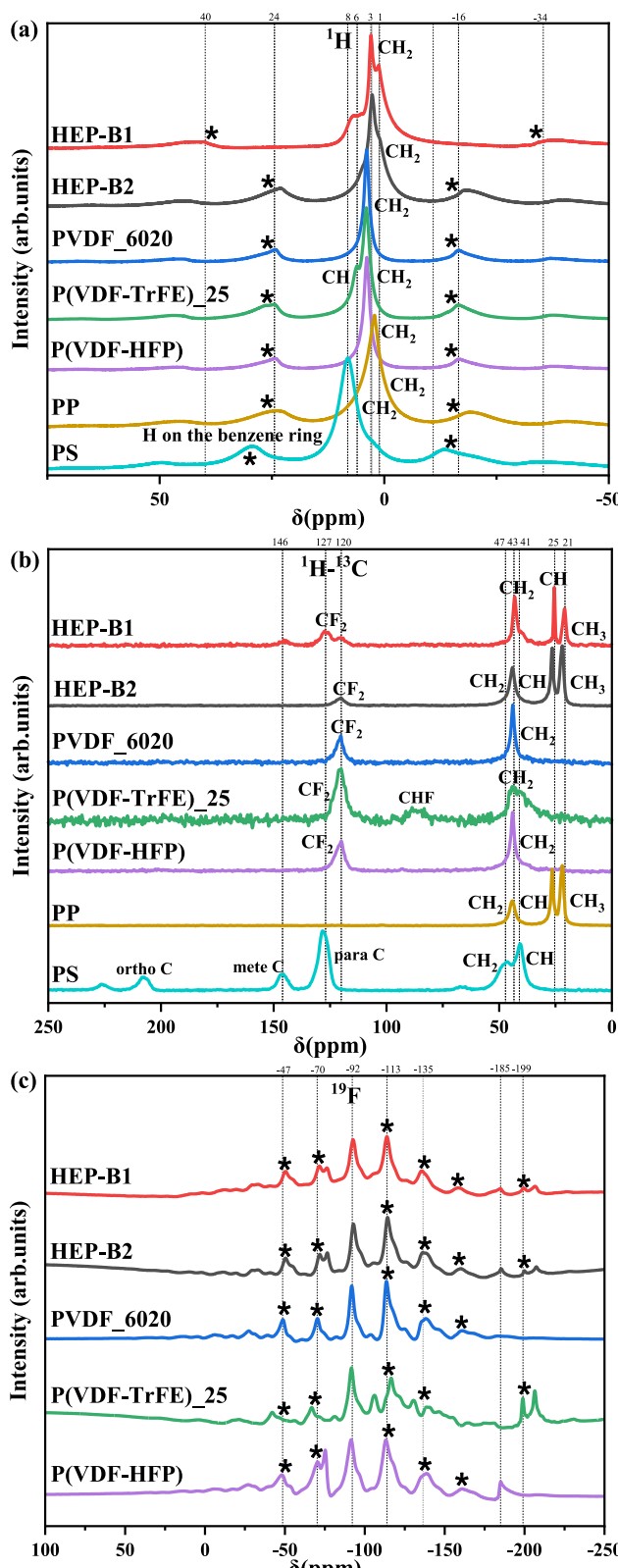

**Fig. 4 | Nuclear Magnetic Resonance (NMR) spectral analysis of HEP-B1 and HEP-B2 polymer blends and individual polymers. a** [1]H NMR spectra. **b** [1]H-[13]C cross-polarization spectra. **c** [19]F NMR spectra.

## Temperature dependence of dielectric properties

The dielectric properties of the individual and blended polymers as a function of temperature at 1 kHz are presented in Fig. 5. The complete set of data for a range of frequencies is presented in Supplementary

Fig. S4 in the Supplementary Information. The non-polar polymers, PP and PS, have a low dielectric constant (~3) and loss tangent (<0.05), and are temperature independent up until above 150 °C, with their permittivity dominated by their electronic polarisation. The polar polymers, PVDF, P(VDF-TrFE), P(VDF-HFP) show more complex behaviour. PVDF and P(VDF-HFP) at room temperature and above have a relatively high (>10) and temperature independent dielectric constant until above 150 °C. Below room temperature their dielectric constant drops rapidly as the temperature approaches the glass transition temperature, which is evidenced by a peak in the dielectric loss tangent; PVDF occurring at −13 °C; P(VDF-TrFE) at −2 °C; and P(VDF-HFP) at −18 °C at a frequency of 1 kHz. Below $T_g$, the amorphous polymer chains of PVDF and P(VDF-TrFE) are no longer able to rotate in response to the applied electric field, and their dielectric constant (<5) and loss tangent (<0.05) become similar to those of the non-polar polymers, PP and PS, corresponding to the electronic polarisability of the amorphous and crystalline polymer chains dominating the dielectric response. Note that the dielectric constant of P(VDF-HFP) below $T_g$ is higher (~7) than that of the other polymers, including the blended polymers. The P(VDF-TrFE) shows a similar dependence on temperature as PVDF and P(VDF-HFP), except its β- crystalline phase is ferroelectric and its Curie transition at ~120 °C produces a peak in the dielectric constant and loss tangent.

The dielectric constant of the blended film B1 is significantly enhanced at and above room temperature compared to the RoM for the individual polymers, while that of the B2 blended film is similar (Fig. 1). The dielectric constant of B1 and B2 below and above $T_g$ are 5.2–22.4 and 3.3–8.7, respectively (Fig. 5). All of the above observations indicate that above $T_g$, the major contribution to the dielectric response comes from the dipole orientation contribution from the amorphous components in the individual fluorinated polymers and blends. Therefore, increasing the proportion of amorphous phase increases the dielectric constant. This is achieved in the B1 blended film by the addition of PS with a high $T_g$ (~100 °C), which increases the viscosity of the blend, frustrating the de-blending of the polymers, which not only increases the proportion of amorphous phase, but also leads to intimate blending of the different components and increased disorder; this produces an increased inter-polymer chain distance that increases the freedom of rotation of polar nano regions, leading to increased dielectric constant[3,24]. Removal of PS from the blend eliminates this effect, and the properties of the blend B2 are then consistent with the RoM for the blend. However, the change in relative dielectric constant of the B1 polymer blend goes beyond what would be expected from simply increasing the proportion of the amorphous phase. To demonstrate this, consider the following: for all of the individual polymers we can convert the difference (contribution from orientation polarisation) between their low (−150 °C, see insert in Fig. 5 and Table 2) and room temperature experimentally measured dielectric constant into the equivalent values for 100% of the amorphous phase content using the data in Tables 1 and 2; this assumes that the electronic polarisation contribution to the dielectric constant of the individual polymers is similar at low temperature and room temperature, and that it is similar for their crystalline and amorphous phases. For B1, this gives a difference in the dielectric constant from the low to high temperature of 11.0 (dipole orientation contribution estimated for 100% of the amorphous phase for each of the individual polymers, using the crystallinity data in Table 1 and dielectric data in Table 2). Combining this with the average value of the low temperature dielectric constant (electronic contribution to dielectric permittivity) of the five individual polymers, 4.1, gives a total permittivity at room temperature of 15.1, well below the experimental value of 22.4, indicating that the increased dielectric constant of the blend B1 compared to the RoM is not simply due to the increased amorphous phase content of the B1 blended polymer, but additionally due to an increase in the dipole orientation rotation space and density of polar nano regions.

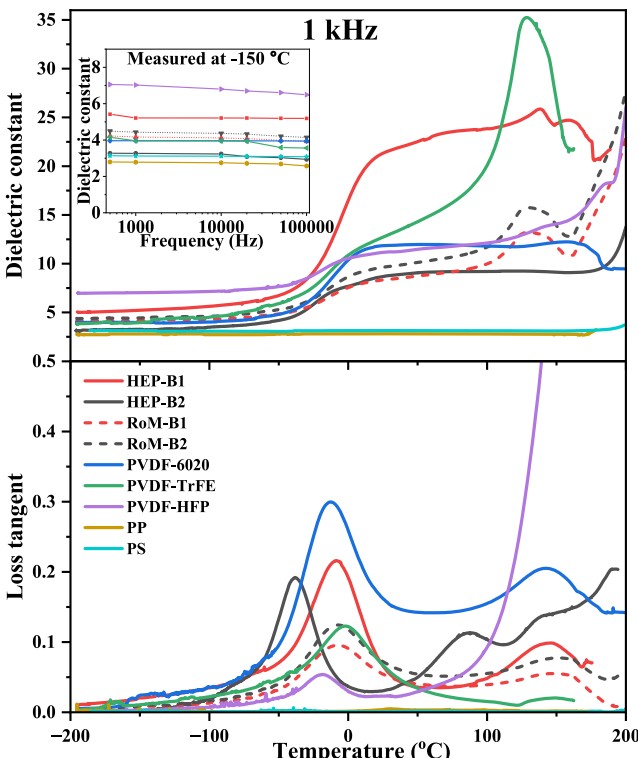

**Fig. 5 | Dielectric properties of HEP-B1 and HEP-B2 polymer blends and individual polymers.** Temperature dependence of dielectric constant and loss tangent at 1 kHz. The experimental data (solid lines) for HEP-B1 and HEP-B2 are compared with the calculated Rule-of-Mixtures (RoM) (dashed lines). Insert, frequency dependence of dielectric constant measured at −150 °C.

**Table 2 | Dielectric data of the blended and individual polymers**

| Name | Dielectric constant at −150 °C and 1 kHz | Dielectric constant and loss at room temperature and 1 kHz [Fig. 1] | |
|---|---|---|---|
| HEP-B1 | 5.2 | 22.4 | 0.005 |
| HEP-B2 | 3.3 | 8.7 | 0.030 |
| PVDF_6020 | 3.9 | 11.8 | 0.020 |
| P(VDF-TrFE)_25 | 3.9 | 12.7 | 0.030 |
| P(VDF-HFP) | 7.0 | 11.1 | 0.040 |
| PP | 2.8 | 2.8 | 0.005 |
| PS | 3.1 | 3.1 | 0.009 |

The microstructures and interactions responsible for the increased amorphous phase content and enhanced dielectric constant of the B1 blend are metastable, as evidenced by the disappearance of the high temperature exothermic peak (-195 °C) in the DSC data upon subsequent cooling and heating (Fig. 2). To further investigate the thermal stability of the dielectric properties, the B1 and B2 polymer blends were annealed at 150 °C for 48 h. This temperature was chosen because it is well above the $T_g$ of all of the polymers and the Curie transition temperature of P(VDF-TrFE), which occurs at -120 °C. It is also significantly higher than the operating temperature of BOPP, which typically functions up to 85 °C[3] or 105 °C[27]. Figure 6a shows a comparison of the room temperature dielectric data before and after annealing, which shows that the dielectric properties of B1 did not change after annealing. While for B2 the dielectric constant remained stable, but the loss tangent increased at the higher frequencies after

annealing. Figure 6b shows the dielectric properties as a function of temperature for typical regions on the same original sample tested as processed and after annealing, and then heated to above 150 °C. The properties of B1 remained stable up to 150 °C, while those of B2 became unstable at >50 °C. This is a significant result because the highly polar polymers with high dielectric constant, like PVDF and its co- and ter-polymers are unstable above 100 °C, which currently limits their use in applications such as link capacitors for electric vehicles.

Figure 7 shows the dielectric properties, plotted as dielectric constant v loss tangent, of the different polymers investigated in this study, with Biaxially-Orientated Polypropylene (BOPP) for comparison[28]. The figure not only highlights the high dielectric constant of the B1 blend, but also the fact that it is achieved with decreasing loss tangent, contrary to what is normally expected, as illustrated by the trend for the other polymers.

## Discussion
We have developed a counter intuitive approach to improve the properties of dielectric polymers using a scalable processing route, the melt blending of multiple immiscible polymers. By an appropriate choice of polymers, the dielectric constant can significantly exceed the rule-of-mixtures value, whilst surprisingly reducing the dielectric loss tangent. The materials also show increased thermal stability up to 150 °C and higher relaxation frequency (>100 MHz), which opens up the possibility of the wider application of dielectric polymers. We provide a consistent model to describe the behaviour based on the higher polarizability of polar nano regions in a highly disordered material. The approach has wide applicability, and by the appropriate choice of polar polymers, their proportions and processing conditions, the microstructures and properties of blended immiscible dielectric properties can be further optimised.

## Methods
### Materials
The individual polymers were selected based on them having similar melt processing temperatures. Table 1 lists useful thermophysical properties, including $T_g$ and crystallite melting temperature. Polyvinylidene fluoride (PVDF) powder with $M_w$ of 670–700 kg mol$^{-1}$ (Solef 6020) was acquired from Solvay S.A. company. Arkema-Piezotech S.A.S (France) supplied P(VDF-TrFE) with a 75:25 molar ratio. Poly(-vinylidene fluoride-co-hexafluoropropylene) (P(VDF-HFP)) with a molecular weight of 400 kg mol$^{-1}$, Polypropylene (PP) beads with $M_w$ of 340 kg mol$^{-1}$ and Polystyrene (PS) with $M_w$ of 200 kg mol$^{-1}$ were purchased from Sigma-Aldrich.

### High entropy polymer film synthesis and preparation
To fabricate the polymer films, a DSM X'plore 15 micro-compounder from Xplore Instruments in Geleen, the Netherlands, was employed via the melt extrusion technique. The extrusion process was conducted at a temperature of 200 °C and a spinning speed of 50 rpm for a duration of 10 minutes to homogenise the melt. The individual polymers were blended in an equimolar-mass ratio. The films were produced using a film die with a slit gap of 150 μm. Then, the blend extruded films were collected by a roller operating at a speed of 150 mm per minute and were simultaneously cooled with water. The thickness of the films was -150 μm.

### Electrical measurement
The temperature and frequency dependence of dielectric properties were obtained measured using an LCR meter (4284 A, Agilent, Santa Clara, CA) and a Precision Impedance Analyzer (Agilent 4294, USA) at 1 V (minimum confidence value for loss 0.005), respectively. The temperature dependence of the dielectric properties was measured over a wide temperature range of −200 to 200 °C. It is difficult to compensate the measurement of loss tangent over the full range. The

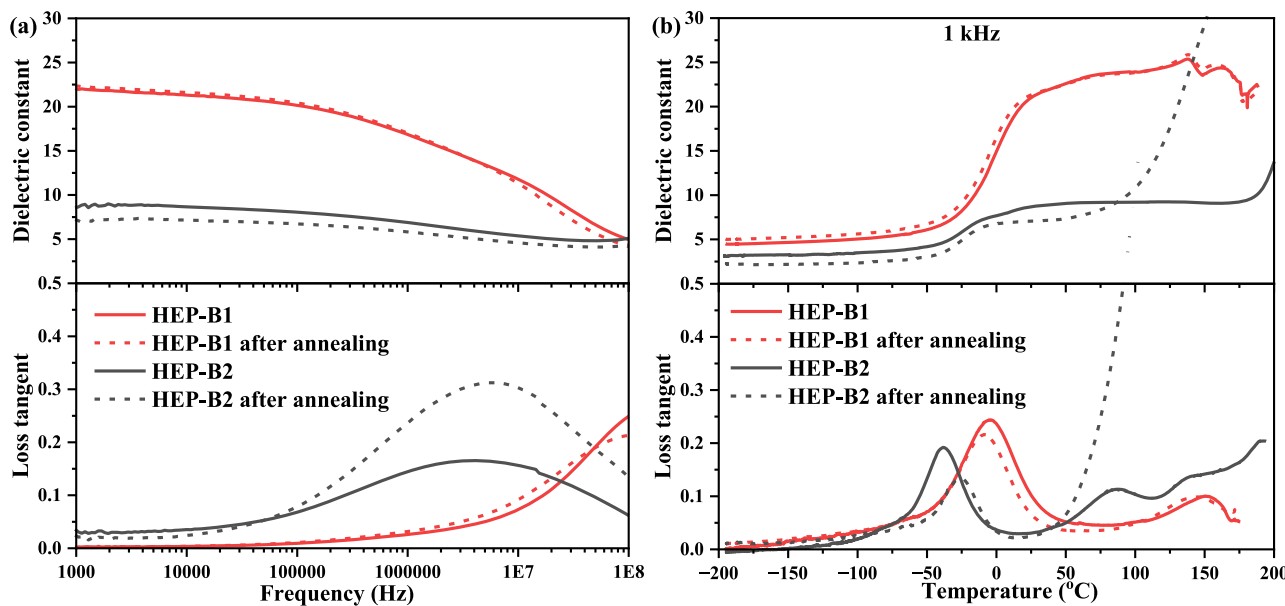

**Fig. 6 | Frequency and temperature-dependent dielectric properties of HEP-B1 and HEP-B2 polymer blends before and after Annealing at 150 °C for 48 h.**
**a** Dielectric constant and loss tangent as a function of frequency. **b** Temperature-dependent dielectric constant and loss tangent at 1 kHz.

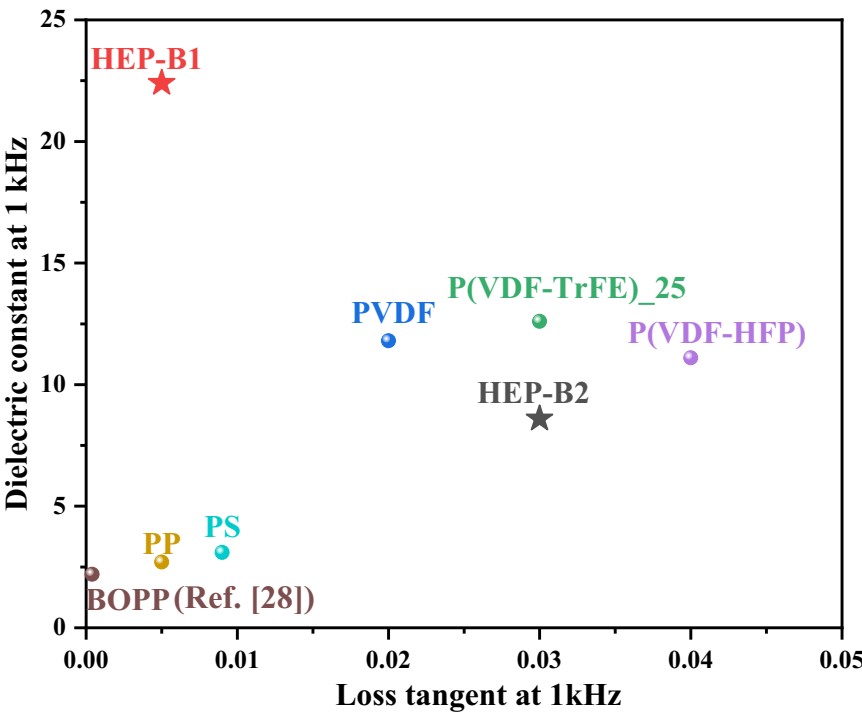

**Fig. 7 | Dielectric performance of HEP-B1 and HEP-B2 polymer blends at 1 kHz.** The plot illustrates the relationship between the dielectric constant and the loss tangent at 1 kHz for two polymer blends, HEP-B1 and HEP-B2. Comparative data from existing literature for BOPP are shown for ref. 28.

results are therefore indicative of the trend in the loss tangent, and very useful for identifying $T_g$. Accurate room temperature values for room temperature loss tangent can be found in the data presented in Fig. 1b, Fig. 6a and Table 2.

### Microstructure characterization

Wide-angle X-ray diffraction (WAXD) was utilized to observe the crystalline structural changes. Cu/Kα radiation with λ = 0.15148 nm was used. The XRD measurements were conducted using an X'Pert Pro instrument manufactured by PANalytical in AImelo, The Netherlands. Each frame for data acquisition had a time of 15 s. The sample-to-

detector distance for the WAXD measurements was set at 207 mm. The crystallinity of the films was calculated from the following equation based on XRD patterns.

$$crystallinity(\chi_c) = \frac{I_c}{I_c + I_a} \times 100\% \tag{4}$$

where, $I_c$ and $I_a$ represent the integrated area of crystalline intensity and amorphous intensity of the XRD diffraction peaks, respectively[29]. Peak separation and area calculation were all done the JADE software.

The microstructure of the films was investigated using polarized optical microscopy (POM) (Olympus BX 60 upright compound microscope, U-CMAD3, Japan).

A Fourier transform infrared spectrometer (FTIR) with the ATR mode (Tensor 27, Optik Bruker GmbH, Germany) was utilized to characterise the crystalline phases.

Raman microscope (Renishaw inViaTM, UK) with a 785 nm laser, 1200 mm⁻¹ line grating, and 50x objective lens (Renishaw, UK) was used to measure Raman spectra between 1750 and 150 $cm^{-1}$.

## Thermal analysis

Differential scanning calorimeter (DSC) (TA, DSC 25, Asse, Belgium) was used for thermal analysis and crystallinity calculations, with a heating rate of 5 °C min⁻¹ and a cooling rate of 5 °C min⁻¹. The overall weight of the individual polymers and polymer blends was the same, 3 mg. The DSC results of the extruded blend film and individual components are illustrated in Fig. 2, with the first heating scan presented in Fig. 2a. The crystallinity ($\chi_c$) was calculated using the formula $(\Delta H_m / \Delta H_{m-100\%}) \times 100\%$.

## Solid-NMR

$^1H$, $^1H$–$^{13}C$, $^{19}F$ Magnetic Resonance Spectroscopy (NMR) measurement was performed on a Bruker Advance AV300 (300 MHz) spectrometer (Illinois, IL, USA). The instrumental settings were configured as follows: $^1H$ NMR: MAS = 12 kHz, 2.5 mm probe, d1 = 1 s. $^1H$-$^{13}C$ NMR: MAS = 12 kHz, 4 mm probe, d1 = 1 s. $^{19}F$ NMR: MAS = 12 kHz, 2.5 mm probe, d1 = 1 s.

## Data availability

The data supporting the findings of this study are available in the paper and the Supplementary Information. Any other relevant data are also available from the corresponding authors upon request.

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

## Acknowledgements

Xin Qi (CSC No.: 202106020045) would like to thank the China Scholarship Council for the financial support. MJR would also like to acknowledge the Engineering and Physical Sciences Research Council (EPSRC) for their support of his research on high entropy materials (EP/Y020936/1).

## Author contributions

X.Q. designed the experiments and performed the synthesis, material characterization, dielectric measurements, and analysis. X.H. performed the XRD analysis. N.K. conducted the solid-state NMR measurements. B.D. helped to process the films. A.P. contributed to the analysis of the XRD and spectroscopy data. H.Y. interpreted the dielectric data. E.B. guided the choice of polymers and optimisation of the processing. D.P. interpreted the characterisation data. M.J.R. conceived the original idea, wrote the original draft, and supervised the research.

## Competing interests

The authors declare no competing interests.
