## [Peer Review file · Nature Communications]

High Entropy Engineered Polymer Blends with Enhanced Dielectric Properties and High Temperature Stability

Corresponding Author: Professor Michael Reece

Version 0:

Reviewer comments:

Reviewer #1

(Remarks to the Author)

This is an interesting work. The authors reported a so-called high-entropy polymer blend by carefully selecting the melt-blended five polymers, showing a greatly enhanced dielectric response far beyond the values predicted by the mixture rules. Such behavior is attributed to the presence of PS, which improves the amorphous content, free volume, and reduces the size of the polar nano regions of the other four polar polymers. This work holds promise as a valuable reference for the development of high-k melt-processable polymer films. The experimental section is detailed enough, facilitating reproducibility. The findings are thoroughly discussed, with conclusions well-supported by experimental evidence. This work merits to be published in a prestigious journal after clarification of certain points.

1. The effect of PS content on the augmentation of permittivity seems saturated above 15%. Pls give some explanation.
2. How about the entropy (disorder) effect in the reported system? In other words, if mixing PS with the other three or two polymers, does the phenomenon still appear?
3. The kinetically trapped polymer blends during cooling during melt extrusion are actually metastable, so are the dielectric properties stable as the B1 or B2 suffer from cycled heating and cooling?
4. LINE 116. polar polymers than "B1".
5. Figure 7, changing the imaginary part into loss tangent data will be more convincing to show that the enhanced permittivity in B1 is not accompanied by an increased loss tangent.
6. Figure 1, changing the Y axis into log format to separate the loss curves at low frequencies, so that the readers will be able to evaluate if the loss tangent values are acceptable for real applications.

Reviewer #2

(Remarks to the Author)

Very interesting work on melt blending of multiple immiscible polymers to enhance the dielectric properties; exceeding that predicted by simple rule of mixtures. Analysis by a range of techniques. Well written and logical - the methodology is therefore sound. The high permittivity can benefit a number of applications, as outlined in the paper.

The blends B1 and B2 could be more clearly introduced to the reader early in the Results section after the introduction, I had to jump to later in the paper to understand.

Inset figures have small text. E.g. fig 5.

The mechanism image Fig 1a is ok for mixing, but it does not make clear how it leads to a high permittivity (greater than the rule of mixtures). It would be of interest to consider how to demonstrate this, or make make clear, by a new image.

Figure 7 is excellent showing high permittivity and low loss. Could a Fig 7(b), as an additional image, somehow have some similar data, but also indicating the temperature stability (as this is something in the title also). Not vital, but would be a good addition as it would need data from all the materials.

Very interesting approach.

(Remarks to the Author)

Over the past five years, the concept of high-entropy polymer blends has emerged as a novel strategy to overcome the typical immiscibility challenges faced in polymer systems. This concept, inspired by the more extensively studied high-entropy alloys and ceramics, has provided a basis for exploring similar behavior in polymer systems. To qualify as a high-entropy polymer blend, the system should contain at least five different components (as in the present case), with one of them often providing a kinetic barrier to prevent the spontaneous demixing of the other components. Currently, it is fair to say that the field of high-entropy polymer blends remains at an early stage, as evidenced by the limited number of studies dedicated to their characterization and experimental investigation. To date, most of the literature has primarily concentrated on understanding and confirming their phase behavior, rather than investigating their practical applications. However, this trend has shifted notably over the past year, with a significant increase in recent reports where the primary motivation for developing such systems is to improve performance across various applications. In this context, the present work, led by Reece, fits well within the topic, as it seeks to bring this concept into the field of polymer dielectrics, an area that has garnered remarkable attention in the past decade due to our current energy landscape. However, even when I consider that the idea overcomes the novelty barrier successfully, the work must be significantly improved prior to being considered for publication. There are a lot of result-discussion relations that appear to be unclear. Below you will find some questions and suggestions that could help to improve the understanding of the present work.

1) The authors must explain how was the thinking process for selecting the polymers used in the fabrication of the blends. In the introduction section they address why they select PVDF and P(VDF-TrFE), and they also give an idea of why PS. But ¿Why PP and P(VDF-HFP)? I assume that there is an important reason rather than just because they have similar melting temperatures, as they stated. Answering this, could help future readers to understand how to select polymer to afford this type of polymer blends.

2) In the abstract section, the authors mentioned that specimens retain low dielectric loss (It should be loss factor or lost tangent). However, Figure 6 clearly shows that at 25 °C, the $\tan(\delta)$ value for HEP-B1 is above 0.5, which is remarkably high compared to most of the polymer dielectrics reported to date (10.1021/acs.macromol.6b02669). Therefore, the dissipative character of these specimens falls notably outside the regimen for being considered as low-dissipative materials.

3) In the abstract section, the authors mention that the thermal stability of the obtained systems is marked by an upper limit of 150 °C, which opens up the possibility of the wider application of dielectric polymers. This is not totally accurate, and indeed, based on this value, the thermal stability of these systems should not be considered enough for most of dielectric applications, particularly those mentioned at the beginning in the introduction section. The following ref (10.1146/annurev-matsci-070317-124435) could help the authors to discuss better this results.

4) In the introduction section, they highlight the issues commonly associated with PFAS-based materials. What is the purpose of mentioning this when they still rely on them to produce the materials for this work?

5) Lines 76-78 contain Tg values accompanied by frequency values. As far as I understand, the Tg values recorded in this work came from DSC analysis. So, what is indicating the frequency value? Did they report the values coming from BDS analysis? If so, were these values extracted from $\tan(\delta)$ maxima?

6) In line 90, the authors mention a "... uniform and translucent appearance." However, Figure S1 clearly shows that both blends exhibit a noticeable degree of opacity, particularly in the case of HEP-B1, which is the most relevant sample. This discrepancy should be addressed in the manuscript, especially considering that the authors also propose in the abstract the presence of "polar nano regions" in the system where demixing is more strongly suppressed. If such regions were indeed on the nanoscale, one would expect reduced light scattering and therefore lower opacity in HEP-B1.

7) Line 98 it is mentioned dielectric loss(ϵ''), and should be loss tangent or loss factor ($\tan(\delta)$).

8) The analysis of the loss tangent isotherms (at room temperature) could be significantly improved by investigating the origin of the relaxation observed. It is clear that this transition arises from the presence of fluorinated components, as it is seen in each individual specimen. However, similarly to how the effect of PS was studied (Figure 1c), it would be advantageous to examine the dielectric response of an equi-proportioned mixture of the three fluorinated polymers to see if this contributes to the relaxation position on the frequency scale (are they miscible between each other? Authors should provide a $\tan(\delta)$ isotherm of this mixture). Additionally, the origin of this relaxation should be clarified, the authors ascribe this transition to dipolar motions but, are these motions coming from local movements, or are they coupled with long-segmental chain movements?. It appears to correspond to an α -relaxation process. In this context, the fact that the relaxation becomes faster in blend B1 could potentially support the authors' hypothesis regarding a confinement effect. Providing a more detailed discussion on this point would strengthen the interpretation.

9) the manuscript increasingly relies on the concept of "polar nano regions" to explain the observed behavior. However, no direct evidence is provided to support their presence. How can the authors be confident in this assumption? If such regions are central to the proposed mechanism, additional experimental validation should be included.

10) In line 181, the authors state that "vibrational spectroscopy is unable to provide direct evidence for the mechanism responsible for the enhanced dielectric response." However, it is well established that vibrational (or atomic) polarization mechanisms contribute only marginally to the overall dielectric constant in polymer materials, typically accounting for about 20–30% of the electronic polarization contribution, based on previous reports. Given the remarkably high dielectric constants reported in this work, particularly for HEP-B1, could the authors clarify why vibrational spectroscopy is discussed in this context? What specific role do they expect it to play in elucidating the dielectric mechanism? As currently stated, the rationale for mentioning vibrational spectroscopy in this context is unclear.

11) Can the authors provide references for the information provided in lines 183 and 184?

12) Instead of NMR, authors should indicate solid-NMR

13) The authors should discuss better why the loss factor of B2 presents such an abrupt change after the annealing process. How the absence of PS is contributing to this?

14) Figure 7 should consider the loss factor ($\tan \delta$) instead of dielectric loss, particular if they want to include BOOP in the

discussion.

15) Just a detail. In thermal analysis, line 336, the author should remove (differential scanning calorimetry)

Version 1:

Reviewer comments:

Reviewer #1

(Remarks to the Author)

The authors fully addressed the reviewers' concerns, and now the manuscript can be published.

Reviewer #2

(Remarks to the Author)

Good response to referees and the paper is improved. The replies to my own comments are clear.

Reviewer #3

(Remarks to the Author)

The authors have properly addressed the comments and suggestions.

Reply to Reviewer Comments

We thank the reviewers for their supportive and useful comments. Our responses to the comments are given below in blue font, and the associated changes in the revised paper are also highlighted in blue font. By addressing the comments from the reviewers, we feel that the quality of the paper has been significantly improved.

Reviewer #1 (Remarks to the Author):

This is an interesting work. The authors reported a so-called high-entropy polymer blend by carefully selecting the melt-blended five polymers, showing a greatly enhanced dielectric response far beyond the values predicted by the mixture rules. Such behavior is attributed to the presence of PS, which improves the amorphous content, free volume, and reduces the size of the polar nano regions of the other four polar polymers. This work holds promise as a valuable reference for the development of high-k melt-processable polymer films. The experimental section is detailed enough, facilitating reproducibility. The findings are thoroughly discussed, with conclusions well-supported by experimental evidence. This work merits to be published in a prestigious journal after clarification of certain points.

We thank the reviewer for their positive and useful comments, which we respond to point-by-point below.

1. The effect of PS content on the augmentation of permittivity seems saturated above 15%. Pls give some explanation.

We agree with the reviewer's observation; there is a decreasing effect with increasing PS content at about 15%. It is important to note that PS has a relatively small dipole moment, and thus makes a relatively minor direct contribution to the dielectric response. As a result, its inclusion alone dilutes the dielectric response. The following text has been added to the paper on page 5:

"It can be seen in **Figure 1 (c)** that at about 15 mass% PS content, the enhancement of permittivity saturates, and the effect of PS on the dielectric properties begins to decrease. Once the beneficial effects of PS addition are optimised, further incorporation results in a dilution of the dielectric constant of the B1 blend."

2. How about the entropy (disorder) effect in the reported system? In other words, if mixing PS with the other three or two polymers, does the phenomenon still appear?

This is an interesting question and a good suggestion for us and others to investigate in future work. It would help to further elucidate the effects of PS and multiple polymer components.

3. The kinetically trapped polymer blends during cooling during melt extrusion are actually metastable, so are the dielectric properties stable as the B1 or B2 suffer from cycled heating and cooling?

We agree that the blends are metastable, and the question is to what temperature are they kinetically stable? The dielectric data after thermal annealing in Figure 6 shows that B1 is stable after a single thermal cycle up to 150 °C, while B2 becomes unstable after annealing at 150 °C and its dielectric properties degrade. At this stage we have no results on the effect of cyclic heating-cooling. However, on page 11 of the original submitted version of the paper we comment on the associated exothermic peaks in the DSC – first heating data, which appear at an even higher temperature, ~195 °C:

“The microstructures and interactions responsible for the increased amorphous phase content and enhanced dielectric permittivity of the B1 blend are metastable, as evidenced by the disappearance of the high temperature exothermic peak (~195 °C) in the DSC data upon subsequent cooling and heating (Figure 2). To further investigate the thermal stability of the dielectric properties, the B1 and B2 polymer blends were annealed at 150 °C for 48 hours.”

4. LINE 116. polar polymers than “B1”.

Corrected.

5. Figure 7, changing the imaginary part into loss tangent data will be more convincing to show that the enhanced permittivity in B1 is not accompanied by an increased loss tangent.

Two of the reviewers have made this point. We have therefore replotted Figure 7 in the way they suggested.

6. Figure 1, changing the Y axis into log format to separate the loss curves at low frequencies, so that the readers will be able to evaluate if the loss tangent values are acceptable for real applications.

This is a good suggestion. Instead, we have provided data at the lowest frequency of 1 kHz in Table 2.

Reviewer #2 (Remarks to the Author):

Very interesting work on melt blending of multiple immiscible polymers to enhance the dielectric properties; exceeding that predicted by simple rule of mixtures. Analysis by a range of techniques. Well written and logical - the methodology is therefore sound. The high permittivity can benefit a number of applications, as outlined in the paper.

We thank the reviewer for their positive comments. We also appreciate their useful comments, which we address point-by-point below.

The blends B1 and B2 could be more clearly introduced to the reader early in the Results section after the introduction, I had to jump to later in the paper to understand.

Inset figures have small text. E.g. fig 5.

This is a helpful point to improve the readability of the paper. We have added this text at the start of the Results section on page 4:

“...B1 (PVDF, P(VDF-TrFE), P(VDF-HFP, PP, PS) and B2 (PVDF, P(VDF-TrFE), P(VDF-HFP, PP)...”

The mechanism image Fig 1a is ok for mixing, but it does not make clear how it leads to a high permittivity (greater than the rule of mixtures). It would be of interest to consider how to demonstrate this, or make clear, by a new image.

We agree with this comment. At this stage, we can simply explain the main effects in terms of PS increasing the amorphous phase content, free volume, and reducing the size of the polar nano regions of the other four polar polymers. The main purpose of Figure 1 (a) is to help with the first point mentioned by the reviewer, and more clearly introduce the blends.

Figure 7 is excellent showing high permittivity and low loss. Could a Fig 7(b), as an additional image, somehow have some similar data, but also indicating the temperature stability (as this is something in the title also). Not vital, but would be a good addition as it would need data from all the materials.

This would be an excellent approach for a future study, once we have a more comprehensive data base, and are more focused on understanding and optimising the temperature stability. The thermal stability after annealing at 150 °C is demonstrated in Figure 6.

Very interesting approach.

Reviewer #3 (Remarks to the Author):

Over the past five years, the concept of high-entropy polymer blends has emerged as a novel strategy to overcome the typical immiscibility challenges faced in polymer systems. This concept, inspired by the more extensively studied high-entropy alloys and ceramics, has provided a basis for exploring similar behavior in polymer systems. To qualify as a high-entropy polymer blend, the system should contain at least five different components (as in the present case), with one of them often providing a kinetic barrier to prevent the spontaneous demixing of the other components. Currently, it is fair to say that the field of high-entropy polymer blends remains at an early stage, as evidenced by the limited number of studies dedicated to their characterization and experimental investigation. To date, most of the literature has primarily concentrated on understanding and confirming their phase behavior, rather than investigating their practical applications. However, this trend has shifted notably over the past year, with a significant increase in recent reports where the primary motivation for developing such systems is to improve performance across various applications. In this context, the present work, led by Reece, fits well within the topic, as it seeks to bring this concept into the field of polymer dielectrics, an area that has garnered remarkable attention in the past decade due to our current energy landscape. However, even when I consider that the idea overcomes the novelty barrier successfully, the work must be significantly improved prior to being considered for publication. There are a lot of result-discussion relations that appear to be unclear. Below you will find some questions and suggestions that could help to improve the understanding of the present work.

We thank the reviewer for their positive comments about the novelty and timeliness of our work. We also appreciate their constructive and useful comments, which we respond to point-by-point below.

1) The authors must explain how was the thinking process for selecting the polymers used in the fabrication of the blends. In the introduction section they address why they select PVDF and P(VDF-TrFE), and they also give an idea of why PS. But

¿Why PP and P(VDF-HFP)? I assume that there is an important reason rather than just because they have similar melting temperatures, as they stated. Answering this, could help future readers to understand how to select polymer to afford this type of polymer blends.

An important criterion for the choice of all of the polymers was that they have similar optimal melt processing conditions so that they could be co-processed using the same temperature, 220 °C. In addition, the rationale for choosing P(VDF-HFP) is that it is a commonly used functional polymer that has polar structures, which can potentially contribute to high permittivity. The rationale for selecting PP is because it is a widely used non-fluorinated polymer, highlighting the possibility to upcycle waste materials. We have added the following text on page 3:

“The fluorinated polymers are commonly used functional polymers that are highly polar which can possibly contribute to high permittivity, while PP and PS are inexpensive, commonly used and non-fluorinated polymers.”

2) In the abstract section, the authors mentioned that specimens retain low dielectric loss (It should be loss factor or lost tangent). However, Figure 6 clearly shows that at 25 °C, the $\tan(\delta)$ value for HEP-B1 is above 0.5, which is remarkably high compared to most of the polymer dielectrics reported to date (10.1021/acs.macromol.6b02669). Therefore, the dissipative character of these specimens falls notably outside the regimen for being considered as low-dissipative materials.

The dielectric results presented in Figure 6b were recorded for a wide temperature range of -200 to 200 °C. It is difficult to compensate the measurement of loss tangent over the full range based on our home made equipment with its additional cables and connections. For that reason, we have not listed the low temperature loss data in Table 2. More accurate values for room temperature can be found in the data presented in Figure 1 (b) and Table 2, where the value of the loss tangent for B1 at room temperature is 0.005 after proper calibration. To clarify this detail, we have added the following text in the Experimental Details on page 13/14:

“The temperature dependence of the dielectric properties were measured over a wide temperature range of -200 to 200 °C. It is difficult to compensate the measurement of loss tangent over the full range. The results are therefore indicative of the trend in the loss tangent, and very useful for identifying T_g . Accurate room temperature values for room temperature loss tangent can be found in the data presented in Figure 1 (b), Fig 6a and Table 2.”

3) In the abstract section, the authors mention that the thermal stability of the obtained systems is marked by an upper limit of 150 °C, which opens up the possibility of the wider application of dielectric polymers. This is not totally accurate, and indeed, based on this value, the thermal stability of these systems should not be considered enough for most of dielectric applications, particularly those mentioned at

the beginning in the introduction section. The following ref (10.1146/annurev-matsci-070317-124435) could help the authors to discuss better this results.

We chose an annealing temperature of 150 °C because it is significantly higher than the Curie transition of P(VDF-TrFE) at ~120 °C. In the current work we have clearly demonstrated the enhanced thermal stability of the blended materials. The temperature exceeds that of the current best-performing industry material, BOPP, which is only effective up to 85 °C [dx.doi.org/10.1021/jz501831q | J. Phys. Chem. Lett. 2014, 5, 3677–3687] or 105 °C [according to the review paper suggested by the referee - 10.1146/annurev-matsci-070317-124435]. We have added the following text on page 11:

“This temperature was chosen because it is well above the T_g of all of the polymers and the Curie transition temperature of P(VDF-TrFE), which occurs at ~120 °C. It is also significantly higher than the operating temperature of BOPP, which typically functions up to 85 °C^[36] or 105 °C^[37].”

We thank the reviewer for highlighting the review paper, which will be useful for guiding our choice of polymers in future work. We have now cited this paper in the revised version of the paper on page 11 as [37].

4) In the introduction section, they highlight the issues commonly associated with PFAS-based materials. What is the purpose of mentioning this when they still rely on them to produce the materials for this work?

The current work demonstrates the approach using well known polar polymers. On page 2 of the paper we say:

“While we demonstrate this approach using a model system containing fluorinated and non-fluorinated polymers, the same approach can be applied to all non-fluorinated polar polymer blends.”

5) Lines 76-78 contain T_g values accompanied by frequency values. As far as I understand, the T_g values recorded in this work came from DSC analysis. So, what is indicating the frequency value? Did they report the values coming from BDS analysis? If so, were these values extracted from $\tan(\delta)$ maxima?

The T_g values were determined from the dielectric data in Figure 5, hence their dependence on frequency.

6) In line 90, the authors mention a “... uniform and translucent appearance.” However, Figure S1 clearly shows that both blends exhibit a noticeable degree of opacity, particularly in the case of HEP-B1, which is the most relevant sample. This discrepancy should be addressed in the manuscript, especially considering that the authors also propose in the abstract the presence of “polar nano regions” in the

system where demixing is more strongly suppressed. If such regions were indeed on the nanoscale, one would expect reduced light scattering and therefore lower opacity in HEP-B1.

For correctness, we have added to page 4 the following text:

“...and translucent appearance of the individual polymers (Supplementary Information, **Figure S1**). The blended polymers, B1 and B2, are less transparent.”

7) Line 98 it is mentioned dielectric loss(ϵ''), and should be loss tangent or loss factor ($\tan(\delta)$).

Corrected.

8) The analysis of the loss tangent isotherms (at room temperature) could be significantly improved by investigating the origin of the relaxation observed. It is clear that this transition arises from the presence of fluorinated components, as it is seen in each individual specimen. However, similarly to how the effect of PS was studied (Figure 1c), it would be advantageous to examine the dielectric response of an equi-proportioned mixture of the three fluorinated polymers to see if this contributes to the relaxation position on the frequency scale (are they miscible between each other? Authors should provide a $\tan(\delta)$ isotherm of this mixture). Additionally, the origin of this relaxation should be clarified, the authors ascribe this transition to dipolar motions but, are these motions coming from local movements, or are they coupled with long-segmental chain movements?. It appears to correspond to an α -relaxation process. In this context, the fact that the relaxation becomes faster in blend B1 could potentially support the authors' hypothesis regarding a confinement effect. Providing a more detailed discussion on this point would strengthen the interpretation.

These are good suggestions for future work to further elucidate the underlying mechanisms and to optimise the properties. We also intend to extend this work to include GHz-frequency measurements. We do already have relevant published data for the PVDF and P(VDF-TrFE) blend [Ref 20 in the paper]; in this work, there was a smaller enhancement (~50%) of the dielectric constant, compared to the expected value from RoM. However, there was no evident change in the relaxation frequency. This relaxation behaviour is also consistent with the results observed for the B2 blend, which contains all three fluorinated polymers, along with PP, but without PS. The relaxation frequency for the B2 blend, remains similar to that of the individual constituent polymers. These findings highlight the importance of PS in enhancing the dielectric properties of the B1 blend, and in reducing its relaxation time. To highlight these observations, we have added the following text on page 5:

“It should also be noted that in previously reported work on a two-polymer blend of PVDF and P(VDF-TrFE) [20], a modest enhancement (~50%) of the dielectric constant was observed compared the expected value from the RoM. However, there was no noticeable change in the relaxation frequency. This relaxation behaviour is also consistent with the results for the B2 blend, which includes the three fluorinated polymers, along with PP, but excludes PS. The relaxation frequency of the B2 blend, is similar to that of the individual polymers. These observations highlight the critical role of PS in enhancing the dielectric properties of the B1 blend, and in reducing its relaxation time.”

9) the manuscript increasingly relies on the concept of “polar nano regions” to explain the observed behavior. However, no direct evidence is provided to support their presence. How can the authors be confident in this assumption? If such regions are central to the proposed mechanism, additional experimental validation should be included.

As the reviewer helpfully pointed in point 8), “In this context, the fact that the relaxation becomes faster in blend B1 could potentially support the authors’ hypothesis regarding a confinement effect. Providing a more detailed discussion on this point would strengthen the interpretation.” We greatly appreciate this comment, which helps to clarify our interpretation. In response, we have incorporated this in the text on page 5:

“This behaviour could be explained by an increasing confinement of the active orientation polar regions with increasing PS content into smaller polar nano regions.”

10) In line 181, the authors state that “vibrational spectroscopy is unable to provide direct evidence for the mechanism responsible for the enhanced dielectric response.” However, it is well established that vibrational (or atomic) polarization mechanisms contribute only marginally to the overall dielectric constant in polymer materials, typically accounting for about 20–30% of the electronic polarization contribution, based on previous reports. Given the remarkably high dielectric constants reported in this work, particularly for HEP-B1, could the authors clarify why vibrational spectroscopy is discussed in this context? What specific role do they expect it to play in elucidating the dielectric mechanism? As currently stated, the rationale for mentioning vibrational spectroscopy in this context is unclear.

The reviewer raises a valid and clear point here. Our vibrational spectroscopy results support this observation. The main reason for conducting this characterisation is that these are standard techniques for characterising polymers. Furthermore, we hoped they might indirectly offer insights into the microstructures of the blends. However, they did not, which in itself is still informative. We have improved the related text on page 8:

“Considering that the enhanced dielectric response is observed at frequencies below 10^8 Hz (Figure 1 (b)), vibrational spectroscopy, corresponding to much higher frequencies, is unable to provide direct evidence for the mechanism responsible for the enhanced dielectric response. This could explain why the blending behaviour reported here has not been reported previously using conventional vibrational spectroscopic techniques [20-21].”

11) Can the authors provide references for the information provided in lines 183 and 184?

We have added this text with references on page 8:

“This could explain why the blending behaviour reported here has not been reported previously using conventional vibrational spectroscopic techniques [20-21].”

12) Instead of NMR, authors should indicate solid-NMR

Corrected.

13) The authors should discuss better why the loss factor of B2 presents such an abrupt change after the annealing process. How the absence of PS is contributing to this?

We can only speculate that the higher T_g (~100 °C) of PS leads to a significantly higher viscosity of the B1 blend compared to B2 during annealing at 150 °C. We have not specifically addressed this in the paper, as the microstructures of B1 and B2 must differ at the nanoscale, as indicated by their very different dielectric properties and behaviour. Therefore, making a direct comparison of the effect of PS on thermal stability is not straightforward. This is a good topic for future work either by us or others.

14) Figure 7 should consider the loss factor (tan d) instead of dielectric loss, particular if they want to include BOOP in the discussion.

This same point is discussed above for Reviewer 1, point 5), for which our response is repeated here:

Two of the reviewers have made this point. We have therefore replotted Figure 7 in the way they suggest.

15) Just a detail. In thermal analysis, line 336, the author should remove (differential scanning calorimetry)

Corrected.